# Understanding the Role of Adaptivity in Machine Teaching: The Case of Version Space Learners

**Yuxin Chen**[†]    **Adish Singla**[‡]    **Oisin Mac Aodha**[†]
**Pietro Perona**[†]    **Yisong Yue**[†]

[†]Caltech, {chenyux, macaodha, perona, yyue}@caltech.edu,
[‡]MPI-SWS, adishs@mpi-sws.org

## Abstract

In real-world applications of education, an effective teacher adaptively chooses the next example to teach based on the learner's current state. However, most existing work in *algorithmic machine teaching* focuses on the batch setting, where adaptivity plays no role. In this paper, we study the case of teaching consistent, version space learners in an interactive setting. At any time step, the teacher provides an example, the learner performs an update, and the teacher observes the learner's new state. We highlight that adaptivity does not speed up the teaching process when considering existing models of version space learners, such as the "worst-case" model (the learner picks the next hypothesis randomly from the version space) and the "preference-based" model (the learner picks hypothesis according to some global preference). Inspired by human teaching, we propose a new model where the learner picks hypotheses according to some local preference defined by the current hypothesis. We show that our model exhibits several desirable properties, e.g., adaptivity plays a key role, and the learner's transitions over hypotheses are smooth/interpretable. We develop adaptive teaching algorithms, and demonstrate our results via simulation and user studies.

## 1   Introduction

Algorithmic machine teaching studies the interaction between a teacher and a student/learner where the teacher's objective is to find an optimal training sequence to steer the learner towards a desired goal [36]. Recently, there has been a surge of interest in machine teaching as several different communities have found connections to this problem setting: (i) machine teaching provides a rigorous formalism for a number of real-world applications including personalized educational systems [35], adversarial attacks [24], imitation learning [6, 14], and program synthesis [18]; (ii) the complexity of teaching ("Teaching-dimension") has strong connections with the information complexity of learning ("VC-dimension") [9]; and (iii) the optimal teaching sequence has properties captured by new models of interactive machine learning, such as curriculum learning [4] and self-paced learning [25].

In the above-mentioned applications, adaptivity clearly plays an important role. For instance, in automated tutoring, adaptivity enables personalization of the content based on the student's current knowledge [31, 33, 17]. In this paper, we explore the *adaptivity gain* in algorithmic machine teaching, i.e., how much speedup a teacher can achieve via adaptively selecting the next example based on the learner's current state? While this question has been well-studied in the context of active learning and sequential decision making [15], the role of adaptivity is much less understood in algorithmic machine teaching. A deeper understanding would, in turn, enable us to develop better teaching algorithms and more realistic learner models to exploit the adaptivity gain.

We consider the well-studied case of teaching a consistent, version space learner. A learner in this model class maintains a version space (i.e., a subset of hypotheses that are consistent with the examples received from a teacher) and outputs a hypothesis from this version space. Here, a hypothesis can be viewed as a function that assigns a label to any unlabeled example. Existing work has studied this class of learner model to establish theoretical connections between the information complexity of teaching vs. learning [13, 37, 11]. Our main objective is to understand, when and by how much, a teacher can benefit by adapting the next example based on the learner's current hypothesis. We compare two types of teachers: (i) an *adaptive teacher* that observes the learner's hypothesis at every time step, and (ii) a *non-adaptive teacher* that only knows the initial hypothesis of the learner and does not receive any feedback during teaching. The non-adaptive teacher operates in a batch setting where the complete sequence of examples can be constructed before teaching begins.

Inspired by real-world teaching scenarios and as a generalization of the global "preference-based" model [11], we propose a new model where the learner's choice of next hypothesis $h'$ depends on some *local* preferences defined by the current hypothesis $h$. For instance, the local preference could encode that the learner prefers to make smooth transitions by picking a consistent hypothesis $h'$ which is "close" to $h$. Local preferences, as seen in Fig. 1, are an important aspect of many machine learning algorithms (e.g., incremental or online learning algorithms [27, 28]) in order to increase robustness and reliability. We present results in the context of two different hypotheses classes, and show through simulation and user studies that adaptivity can play a crucial role when teaching learners with local preferences.

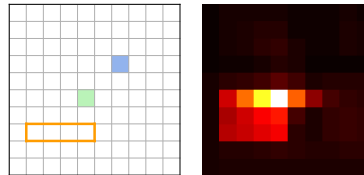

Figure 1: Local update preference. Users were asked to update the position of the orange rectangle so that green cells were inside and blue ones outside. The heatmap on the right displays the updated positions.

## 2 Related Work

**Models of version space learners** Within the model class of version space learners, there are different variants of learner models depending upon their anticipated behavior, and these models lead to different notions of teaching complexity. For instance, (i) the "worst-case" model [13] essentially assumes nothing and the learner's behavior is completely unpredictable, (ii) the "cooperative" model [37] assumes a smart learner who anticipates that she is being taught, and (iii) the "preference-based" model [11] assumes that she has a global preference over the hypotheses. Recently, some teaching complexity results have been extended beyond version space learners, such as Bayesian learners [34], probabilistic/randomized learners [30, 3], learners implementing an optimization algorithm [22], and for iterative learning algorithms based on gradient updates [23]. Here, we focus on the case of version space learners, leaving the extension to other types of learners for future work.

**Batch vs. sequential teaching** Most existing work on algorithmic machine teaching has focused on the batch setting, where the teacher constructs a set of examples and provides it to the learner at the beginning of teaching [13, 37, 11, 7]. There has been some work on sequential teaching models that are more suitable for understanding the role of adaptivity. Recently, [23] studied the problem of iteratively teaching a gradient learner by providing a sequence of carefully constructed examples. However, since the learner's update rule is completely deterministic, a non-adaptive teacher with knowledge of the learner's initial hypothesis $h^0$ would behave exactly the same as an adaptive teacher (i.e., the adaptivity gain is zero). [3] studied randomized version-space learners with limited memory, and demonstrated the power of adaptivity for a specific class of hypotheses. Sequential teaching has also been studied in the context of crowdsourcing applications by [19] and [29], empirically demonstrating the improved performance of adaptive vs. non-adaptive teachers. However, these approaches do not provide any theoretical understanding of the adaptivity gain as done in our work.

**Incremental learning and teaching** Our learner model with local preferences is quite natural in real-world applications. A large class of iterative machine learning algorithms are based on the idea of incremental updates which in turn is important for the robustness and generalization of learning [27, 28]. From the perspective of a human learner, the notion of incremental learning aligns well with the concept of the "Zone of Proximal Development (ZPD)" in the educational research and psychology literature [32]. The ZPD suggests that teaching is most effective when focusing on a task *slightly* beyond the current abilities of the student as the human learning process is inherently

incremental. Different variants of learner model studied in the cognitive science literature [21, 5, 26] have an aspect of incremental learning. For instance, the "win stay lose shift" model [5] is a special case of the local preference model that we propose in our work. Based on the idea of incremental learning, [2] studied the case of teaching a variant of the version space learner when restricted to incremental learning and is closest to our model with local preferences. However, there are two key differences in their model compared to ours: (i) they allow learners to select inconsistent hypotheses (i.e., outside the version space), (ii) the restricted movement in their model is a hard constraint which in turns means that teaching is not always feasible – given a problem instance it is NP-Hard to decide if a given target hypothesis is teachable or not.

## 3   The Teaching Model

We now describe the teaching domain, present a generic model of the learner and the teacher, and then state the teacher's objective.

### 3.1   The Teaching Domain

Let $\mathcal{X}$ denote a ground set of unlabeled examples, and $\mathcal{Y}$ denote the set of possible labels that could be assigned to elements of $\mathcal{X}$. We denote by $\mathcal{H}$ a finite class of hypotheses, each element $h \in \mathcal{H}$ is a function $h : \mathcal{X} \to \mathcal{Y}$. In this paper, we will only consider boolean functions and hence $\mathcal{Y} = \{0, 1\}$. In our model, $\mathcal{X}$, $\mathcal{H}$, and $\mathcal{Y}$ are known to both the teacher and the learner. There is a *target hypothesis* $h^* \in \mathcal{H}$ that is known to the teacher, but not the learner. Let $\mathcal{Z} \subseteq \mathcal{X} \times \mathcal{Y}$ be the ground set of labeled examples. Each element $z = (x_z, y_z) \in \mathcal{Z}$ represents a labeled example where the label is given by the target hypothesis $h^*$, i.e., $y_z = h^*(x_z)$. Here, we define the notion of *version space* needed to formalize our model of the learner. Given a set of labeled examples $Z \subseteq \mathcal{Z}$, the version space induced by $Z$ is the subset of hypotheses $\mathcal{H}(Z) \in \mathcal{H}$ that are consistent with the labels of all the examples, i.e., $\mathcal{H}(Z) := \{h \in \mathcal{H} : \forall z = (x_z, y_z) \in Z, h(x_z) = y_z\}$.

### 3.2   Model of the Learner

We now introduce a generic model of the learner by formalizing our assumptions about how she adapts her hypothesis based on the labeled examples received from the teacher. A key ingredient of this model is the *preference function* of the learner over the hypotheses as described below. As we show in the next section, by providing specific instances of this preference function, our generic model reduces to existing models of version space learners, such as the "worst-case" model [13] and the global "preference-based" model [11].

Intuitively, the preference function encodes the learner's transition preferences. Consider that the learner's current hypothesis is $h$, and there are two hypotheses $h'$, $h''$ that they could possibly pick as the next hypothesis. We want to encode whether the learner has any preference in choosing $h'$ or $h''$. Formally, we define the preference function as $\sigma : \mathcal{H} \times \mathcal{H} \to \mathbb{R}_+$. Given current hypothesis $h$ and any two hypothesis $h', h''$, we say that $h'$ is preferred to $h''$ from $h$, iff $\sigma(h'; h) < \sigma(h''; h)$. If $\sigma(h'; h) = \sigma(h''; h)$, then the learner could pick either one of these two.

The learner starts with an initial hypothesis $h_0 \in \mathcal{H}$ before receiving any labeled examples from the teacher. Then, the interaction between the teacher and the learner proceeds in discrete time steps. At any time step $t$, let us denote the labeled examples received by the learner up to (but not including) time step $t$ via a set $Z_t$, the learner's version space as $\mathcal{H}_t = \mathcal{H}(Z_t)$, and the current hypothesis as $h_t$. At time step $t$, we model the learning dynamics as follows: (i) the learner receives a new example $z_t$; and (ii) the learner updates the version space $\mathcal{H}_{t+1}$, and picks the next hypothesis based on the current hypothesis $h_t$, version space $\mathcal{H}_{t+1}$, and the preference function $\sigma$:

$$h_{t+1} \in \{h \in \mathcal{H}_{t+1} : \sigma(h; h_t) = \min_{h' \in \mathcal{H}_{t+1}} \sigma(h'; h_t)\}. \tag{3.1}$$

### 3.3   Model of the Teacher and the Objective

The teacher's goal is to steer the learner towards the target hypothesis $h^*$ by providing a sequence of labeled examples. At time step $t$, the teacher selects a labeled example $z_t \in \mathcal{Z}$ and the learner transitions from the current $h_t$ to the next hypothesis $h_{t+1}$ as per the model described above. Teaching

finishes at time step $t$ if the learner's hypothesis $h_t = h^*$. Our objective is to design teaching algorithms that can achieve this goal in a minimal number of time steps. We study the *worst-case* number of steps needed as is common when measuring the information complexity of teaching [13, 37, 11].

We assume that the teacher knows the learner's initial hypothesis $h_0$ as well as the preference function $\sigma(\cdot; \cdot)$. In order to quantify the gain from adaptivity, we compare two types of teachers: (i) an *adaptive teacher* who observes the learner's hypothesis $h_t$ before providing the next labeled example $z_t$ at any time step $t$; and (ii) a *non-adaptive teacher* who only knows the initial hypothesis of the learner and does not receive any feedback from the learner during the teaching process. Given these two types of teachers, we want to measure the *adaptivity gain* by quantifying the difference in teaching complexity of the *optimal adaptive* teacher compared to the *optimal non-adaptive* teacher.

## 4 The Role of Adaptivity

In this section, we study different variants of the learner's preference function, and formally state the adaptivity gain with two concrete problem instances.

### 4.1 State-independent Preferences

We first consider a class of preference models where the learner's preference about the next hypothesis does not depend on her current hypothesis. The simplest state-independent preference is captured by the "worst-case" model [13], where the learner's preference over all hypotheses is uniform, i.e., $\forall h, h', \sigma(h'; h) = c$, where $c$ is some constant.

A more generic state-independent preference model is captured by non-uniform, global preferences. More concretely, for any $h' \in \mathcal{H}$, we have $\sigma(h'; h) = c_{h'} \ \forall h \in \mathcal{H}$, a constant dependent only on $h'$. This is similar to the notion of the global "preference-based" version space learner introduced by [11].

**Proposition 1** *For the state-independent preference, adaptivity plays no role, i.e., the sample complexities of the optimal adaptive teacher and the optimal non-adaptive teacher are the same.*

In fact, for the uniform preference model, the teaching complexity of the adaptive teacher is the same as the *teaching dimension* of the hypothesis class with respect to teaching $h^*$, given by

$$\mathsf{TD}(h^*, \mathcal{H}) := \min_Z |Z|, \text{ s.t. } \mathcal{H}(Z) = \{h^*\}. \tag{4.1}$$

For the global preference model, similar to the notion of *preference-based teaching dimension* [11], the teaching complexity of the adaptive teacher is given by

$$\min_Z |Z|, \text{ s.t. } \forall h \in \mathcal{H}(Z) \backslash \{h^*\}, \sigma(h; \cdot) > \sigma(h^*; \cdot). \tag{4.2}$$

### 4.2 State-dependent Preferences

In real-world teaching scenarios, human learners incrementally build up their knowledge of the world, and their preference of the next hypothesis naturally depends on their current state. To better understand the behavior of an adaptive teacher under a state-dependent preference model, we investigate the following two concrete examples:

**Example 1 (2-REC)** *$\mathcal{H}$ consists of up to two disjoint rectangles[1] on a grid and $\mathcal{X}$ represents the grid cells (cf. Fig. 1 and Fig. 3a). Consider an example $z = (x_z, y_z) \in \mathcal{Z}$: $y_z = 1$ (positive) if the grid cell $x_z$ lies inside the target hypothesis, and $0$ (negative) elsewhere.*

The 2-REC hypothesis class consists of two subclasses, namely $\mathcal{H}^1$: all hypotheses with one rectangle, and $\mathcal{H}^2$: those with exactly two (disjoint) rectangles. The 2-REC class is inspired by teaching a union of disjoint objects. Here, objects correspond to rectangles and any $h \in \mathcal{H}$ represents one or two rectangles. Furthermore, each hypothesis $h$ is associated with a complexity measure given by the number of objects in the hypothesis. [10] recently studied the problem of teaching a union of disjoint geometric objects, and [1] studied the problem of teaching a union of monomials. Their results show

that, in general, teaching a target hypothesis of lower complexity from higher complexity hypotheses is the most challenging task.

For the 2-REC class, we assume the following local preferences: (i) in general, the learner prefers to transition to a hypothesis with the same complexity as the current one (i.e., $\mathcal{H}^1 \to \mathcal{H}^1$ or $\mathcal{H}^2 \to \mathcal{H}^2$), (ii) when transitioning within the same subclass, the learner prefers small edits, e.g., by moving the smallest number of edges possible when changing their hypothesis, and (iii) the learner could switch to a subclass of lower complexity (i.e., $\mathcal{H}^2 \to \mathcal{H}^1$) in specific cases. We provide a detailed description of the preference function in the extended version of this paper [8].

**Example 2 (LATTICE)** $\mathcal{H}$ *and* $\mathcal{X}$ *both correspond to nodes in a* 2*-dimensional integer lattice of length* $n$. *For a node* $v$ *in the grid, we have an associated* $h_v \in \mathcal{H}$ *and* $x_v \in \mathcal{X}$. *Consider an example* $z_v = (x_{z_v}, y_{z_v}) \in \mathcal{Z}$: $y_{z_v} = 0$ *(negative) if the target hypothesis corresponds to the same node* $v$, *and* 1 *(positive) elsewhere. We consider the problem of teaching with positive-only examples.*

LATTICE class is inspired by teaching in a physical world from positive-only (or negative-only) reinforcements, for instance, teaching a robot to navigate to a target state by signaling that the current location is not the target. The problem of learning and teaching with positive-only examples is an important question with applications to learning languages and reinforcement learning tasks [12, 20]. For the LATTICE class, we assume that the learner prefers to move to a close-by hypothesis measured via $L1$ (Manhattan) distance, and when hypotheses have equal distances we assume that the learner prefers hypotheses with larger coordinates.

**Theorem 2** *For teaching the* 2-REC *class, the ratio between the cost of the optimal non-adaptive teacher and the optimal adaptive teacher is* $\Omega\left(|h_0|/\log|h_0|\right)$, *where* $|h_0|$ *denotes the number of positive examples induced by the learner's initial hypothesis* $h_0$; *for teaching the* LATTICE *class, the difference between the cost of the optimal non-adaptive teacher and the optimal adaptive teacher is* $\Omega(n)$.

In the above theorem, we show that for both problems, under natural behavior of an incremental learner, adaptivity plays a key role. The proof of Theorem 2 is provided in the extended version of this paper [8]. Specifically, we show the teaching sequences for an adaptive teacher which matches the above bounds for the 2-REC and LATTICE classes. We also provide lower bounds for any non-adaptive algorithms for these two classes. Here, we highlight two necessary conditions under which adaptivity can possibly help: (i) preferences are local and (ii) there are ties among the learner's preference over hypotheses. The learner's current hypothesis, combined with the local preference structure, gives the teacher a handle to steer the learner in a controlled way.

## 5 Adaptive Teaching Algorithms

In this section, we first characterize the optimal teaching algorithm, and then propose a non-myopic adaptive teaching framework.

### 5.1 The Optimality Condition

Assume that the learner's current hypothesis is $h$, and the current version space is $H \subseteq \mathcal{H}$. Let $D^*(h, H, h^*)$ denote the minimal number of examples required in the worst-case to teach $h^*$. We identify the following optimality condition for an adaptive teacher:

**Proposition 3** *A teacher achieves the minimal teaching cost, if and only if for all states* $(h, H)$ *of the learner, it picks an example such that*

$$z^* \in \arg\min_z \left( 1 + \max_{h' \in \mathbf{C}(h, H, \sigma, z)} D^*\left(h', H \cap \mathcal{H}(\{z\}), h^*\right) \right)$$

*where* $\mathbf{C}(h, H, \sigma, z)$ *denotes the set of candidate hypotheses in the next round as defined in* (3.1)*, and for all* $(h, H)$*, it holds that*

$$D^*(h, H, h^*) = \min_z \left( 1 + \max_{h' \in \mathbf{C}(h, H, \sigma, z)} D^*\left(h', H \cap \mathcal{H}(\{z\}), h^*\right) \right)$$

| **Algorithm 1** Non-myopic adaptive teaching |
|---|
| **input:** $\mathcal{H}$, $\sigma$, initial $h_0$, target $h^*$. |
| Initialize $t \leftarrow 0$, $\mathcal{H}_0 \leftarrow \mathcal{H}$ |
| **while** $h_t \neq h^*$ **do** |
| $\quad h_t^* \leftarrow \mathsf{Oracle}(h_t, \mathcal{H}_t, h^*)$ |
| $\quad z_{t+1} \leftarrow \mathsf{Teacher}(\sigma, h_t, \mathcal{H}_t, h_t^*)$ |
| $\quad$ Learner makes an update |
| $\quad t \leftarrow t + 1$ |

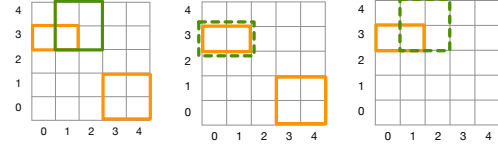

Figure 2: An illustrative example for 2-REC. $h_t$, $h^*$, and $h_t^*$ are represented by the orange rectangles, solid green rectangle and dashed green rectangles, respectively. (Left) The teaching task. (Middle) Sub-task 1. (Right) Sub-task 2.

In general, computing the optimal cost $D^*$ for non-trivial preference functions, including the uniform/global preference, requires solving a linear equation system of size $|\mathcal{H}| \cdot 2^{|\mathcal{H}|}$.

**State-independent preference** When the learner's preference is uniform, $D_u^*(h, H, h^*) = \mathsf{TD}(h^*, H)$ (Eq. 4.1) denotes the set cover number of the version space, which is NP-Hard to compute. A myopic heuristic which gives best approximation guarantees for a polynomial time algorithm (with cost that is within a logarithmic factor of the optimal cost [13]) is given by $\tilde{D}_u(h, H, h^*) = |H|$. For the global preference, the optimal cost $D_g^*(h, H, h^*)$ is given by Eq. (4.2). i.e., the set cover number of all hypotheses in the version space that are more or equally preferred over $h^*$. Similarly, one can also follow the greedy heuristic, i.e., $\tilde{D}_g(h, H, h^*) = |\{h' \in H : \sigma(h'; \cdot) \leqslant \sigma(h^*; \cdot)\}|$ to achieve a logarithmic factor approximation.

**General preference** Inspired by the two myopic heuristics above, we propose the following heuristic for general preference models:

$$\tilde{D}(h, H, h^*) = |\{h' \in H : \sigma(h'; h) \leqslant \sigma(h^*; h)\}| \tag{5.1}$$

In words, $\tilde{D}$ denotes the index of the target hypothesis $h^*$ in the preference vector associated with $h$ in the version space $H$. Notice that for the uniform (resp. global) preference model, the function $\tilde{D}$ reduces to $\tilde{D}_u$ (resp. $\tilde{D}_g$). In the following theorem, we provide a sufficient condition for the myopic adaptive algorithm that greedily minimizes Eq. (5.1) to attain provable guarantees:

**Theorem 4** *Let $h_0 \in \mathcal{H}$ be the learner's initial hypothesis, and $h^* \in \mathcal{H}$ be the target hypothesis. For any $H \subseteq \mathcal{H}$, let $\bar{H}(\{z\}) = \{h' \in H : h'(x_z) \neq y_z\}$ be the set of hypotheses in $H$ which are inconsistent with the teaching example $z \in \mathcal{Z}$. If for all learner's states $(h, H)$, the preference and the structure of the teaching examples satisfy:*

1. *$\forall h_i, h_j \in H, \sigma(h_i; h) \leqslant \sigma(h_j; h) \leqslant \sigma(h^*; h) \implies \sigma(h_j; h_i) \leqslant \sigma(h^*; h_i)$*
2. *$\forall H' \subseteq \bar{H}(\{z\})$, there exists $z' \in \mathcal{Z}$, s.t., $\bar{H}(\{z'\}) = H'$,*

*then, the cost of the myopic algorithm that greedily minimizes[2] (5.1) is within a factor of $2(\log \tilde{D}(h_0, \mathcal{H}, h^*) + 1)$ approximation of the cost of the optimal adaptive algorithm.*

We defer the proof of the theorem to the extended version of this paper [8]. Note that both the uniform preference model and the global preference model satisfy Condition 1. Intuitively, the first condition states that there does not exist any hypothesis between $h$ and $h^*$ that provides a "short-cut" to the target. Condition 2 implies that we can always find teaching examples that ensure smooth updates of the version space. For instance, a feasible setting that fits Condition 2 is where we assume that the teacher can synthesize an example to remove any subset of hypotheses of size at most $k$, where $k$ is some constant.

## 5.2 Non-Myopic Teaching Algorithms

When the conditions provided in Theorem 4 do not hold, the myopic heuristic (5.1) could perform poorly. An important observation from Theorem 4 is that, when $\tilde{D}(h, H, h^*)$ is small, i.e., $h^*$ is close

to the learner's current hypothesis in terms of preference ordering, we need less stringent constraints on the preference function. This motivates adaptively devising intermediate target hypotheses to ground the teaching task into multiple, separate sub-tasks. Such divide-and-conquer approaches have proven useful for many practical problems, e.g., constructing a hierarchical decomposition for reinforcement learning tasks [16]. In the context of machine teaching, we assume that there is an oracle, $\mathsf{Oracle}(h, H, h^*)$ that maps the learner's state $(h, H)$ and the target hypothesis $h^*$ to an intermediate target hypothesis, which defines the current sub-task.

We outline the non-myopic adaptive teaching framework in Algorithm 1. Here, the subroutine $\mathsf{Teacher}$ aims to provide teaching examples that bring the learner closer to the intermediate target hypothesis. As an example, let us consider the 2-REC hypothesis class. In particular, we consider the challenging case where the target hypothesis $h^* \in \mathcal{H}^1$ represents a single rectangle $r^\star$, and the learner's initial hypothesis $h_0 \in \mathcal{H}^2$ has two rectangles $(r_1, r_2)$. Imagine that the first rectangle $r_1$ is overlapping with $r^\star$, and the second rectangle $r_2$ is disjoint from $r^\star$. To teach the hypothesis $h^*$, the first sub-task (as provided by the oracle) is to eliminate the rectangle $r_2$ by providing negative examples so that the learner's hypothesis represents a single rectangle $r_1$. Then, the next sub-task (as provided by the oracle) is to teach $h^*$ from $r_1$. We illustrate the sub-tasks in Fig. 2, and provide the full details of the adaptive teaching algorithm (i.e., $\mathsf{Ada\text{-}R}$ as used in our experiments) in the extended version of this paper [8].

## 6 Experiments

In this section, we empirically evaluate our teaching algorithms on the 2-REC hypothesis class via simulated learners.

### 6.1 Experimental Setup

For the 2-REC hypothesis class (cf. Fig. 3a and Example 1), we consider a grid with size varying from $5 \times 5$ to $20 \times 20$. The ground set of unlabeled teaching examples $\mathcal{X}$ consists of all grid cells. In our simulations, we consider all four possible teaching scenarios, $\mathcal{H}^{1 \rightarrow 1}$, $\mathcal{H}^{1 \rightarrow 2}$, $\mathcal{H}^{2 \rightarrow 1}$, $\mathcal{H}^{2 \rightarrow 2}$, where $i, j$ in $\mathcal{H}^{i \rightarrow j}$ specify the subclasses of the learner's initial hypothesis $h_0$ and the target hypothesis $h^*$. In each simulated teaching session, we sample a random pair of hypotheses $(h_0, h^*)$ from the corresponding subclasses.

**Teaching algorithms**  We consider three different teaching algorithms as described below. The first algorithm, $\mathsf{SC}$, is a greedy set cover algorithm, where the teacher greedily minimizes $\tilde{D}_u = |H|$ (see §5.1). In words, the teacher acts according to the uniform preference model, and greedily picks the teaching example that eliminates the most inconsistent hypotheses in the version space. The second algorithm, denoted by $\mathsf{Non\text{-}R}$ for the class 2-REC, represents the non-adaptive teaching algorithm that matches the non-adaptive lower bounds provided in Theorem 2, with implementation details provided in the extended version of this paper [8]. Note that both $\mathsf{SC}$ and $\mathsf{Non\text{-}R}$ are non-adaptive. The third algorithm, $\mathsf{Ada\text{-}R}$, represents the non-myopic adaptive teaching algorithm instantiated from Algorithm 1. The details of the subroutines $\mathsf{Oracle}$ and $\mathsf{Teacher}$ for $\mathsf{Ada\text{-}R}$ are provided in the extended version of this paper [8]. We note that all teaching algorithms have the same stopping criterion: the teacher stops when the learner reaches the target hypothesis, that is, $h_t = h^*$.

### 6.2 Results

We measure the performance of the teaching algorithms by their teaching complexity, and all results are averaged over 50 trials with random samples of $(h_0, h^*)$.

**Noise-free setting**  Here, we consider the "noise-free" setting, i.e., the learner acts according to the state-dependent preference models as described in §4.2. In Fig. 3b, we show the results for 2-REC class with a fixed grid size $15 \times 15$ for all four teaching scenarios. As we can see from Fig. 3b, $\mathsf{Ada\text{-}R}$ has a consistent advantage over the non-adaptive baselines across all four scenarios. As expected, teaching $\mathcal{H}^{1 \rightarrow 1}$, $\mathcal{H}^{1 \rightarrow 2}$, and $\mathcal{H}^{2 \rightarrow 2}$ is easier, and the non-adaptive algorithms ($\mathsf{SC}$ and $\mathsf{Non\text{-}R}$) perform well. In contrast, when teaching $\mathcal{H}^{2 \rightarrow 1}$, we see a significant gain from $\mathsf{Ada\text{-}R}$ over the non-adaptive baselines. In the worst case, $\mathsf{SC}$ has to explore *all* the negative examples to teach $h^*$, whereas $\mathsf{Non\text{-}R}$ needs to consider all negative examples within the learner's initial hypothesis

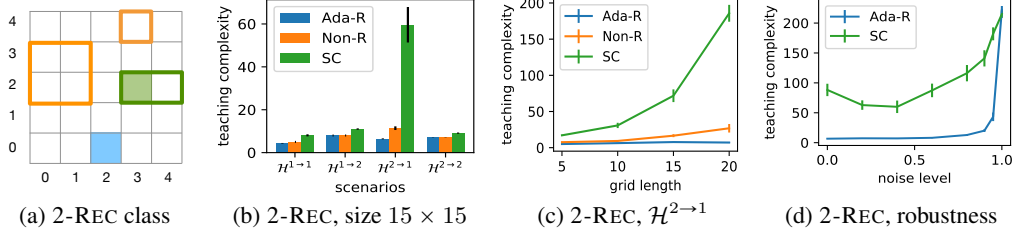

(a) 2-REC class      (b) 2-REC, size $15 \times 15$      (c) 2-REC, $\mathcal{H}^{2 \to 1}$      (d) 2-REC, robustness

Figure 3: Illustration and simulation results for 2-REC. (a) illustrates the 2-REC hypothesis class. The initial hypothesis $h_0 \in \mathcal{H}^2$ is represented by the orange rectangles, and the target hypothesis $h^* \in \mathcal{H}^1$ is represented by the green rectangle. The green and blue cells represent a positive and a negative teaching example, respectively. Simulation results are shown in (b)-(d).

$h_0$ to make the learner jump from the subclass $\mathcal{H}^2$ to $\mathcal{H}^1$. In Fig. 3c, we observe that the adaptivity gain increases drastically as we increase the grid size. This matches our analysis of the logarithmic adaptivity gain in Theorem 2 for 2-REC.

**Robustness in a noisy setting** In real-world teaching tasks, the learner's preference may deviate from the preference $\sigma$ of an "ideal" learner that the teacher is modeling. In this experiment, we consider a more realistic scenario, where we simulate the noisy learners by randomly perturbing the preference of the "ideal" learner at each time step. With probability $1 - \varepsilon$ the learner follows $\sigma$, and with probability $\varepsilon$, the learner switches to a random hypothesis in the version space. In Fig. 3d, we show the results for the 2-REC hypothesis class with different noise levels $\varepsilon \in [0, 1]$. We observe that even for highly noisy learners e.g., $\varepsilon = 0.9$, our algorithm Ada-R performs much better than SC. [3,4]

## 7 User Study

Here we describe experiments performed with human participants from Mechanical Turk using the 2-REC hypothesis class. We created a web interface in order to (i) elicit the preference over hypotheses of human participants, and to (ii) evaluate our adaptive algorithm when teaching human learners.

**Eliciting human preferences** We consider a two-step process for the elicitation experiments. At the beginning of the session (first step), participants were shown a grid of green, blue, or white cells and asked to draw a hypothesis from the 2-REC class represented by one or two rectangles. Participants could only draw "valid" hypothesis which is consistent with the observed labels (i.e., the hypothesis should contain all the green cells and exclude all the blue cells), cf. Fig. 3a. The color of the revealed cells is defined by an underlying target hypothesis $h^*$. In the second step, the interface updated the configuration of cells (either by adding or deleting green/blue cells) and participants were asked to redraw their rectangle(s) (or move the edges of the previously drawn rectangle(s)) which ensures that the updated hypothesis is consistent.

We consider 5 types of sessions, depending on the class of $h^*$ and configurations presented to a participant in the first and the second step. These configurations are listed in Fig. 4a. For instance, the session type in the third row $(\mathcal{H}^2, (1/2), 2)$ means the following: the labels were generated based on a hypothesis $h^* \in \mathcal{H}^2$; in the first step, both subclasses $\mathcal{H}^1$ and $\mathcal{H}^2$ had consistent hypotheses; and in the second step, only the subclass $\mathcal{H}^2$ had consistent hypotheses.

We tested 215 participants, where each individual performed 10 trials on a grid of size $12 \times 12$. For each trial, we randomly selected one of the five types of sessions as discussed above. In Fig. 4a, we see

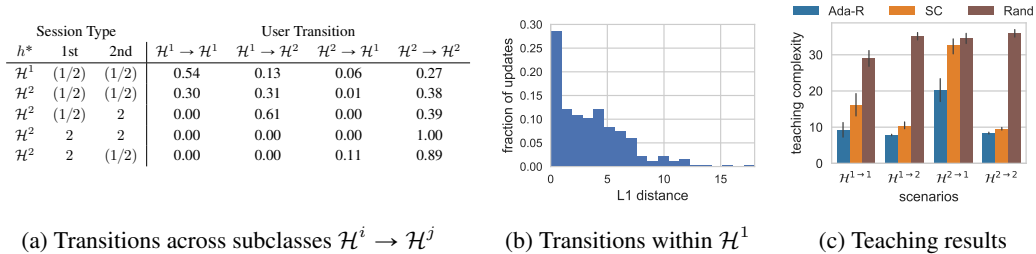

| Session Type | | | User Transition | | | |
|---|---|---|---|---|---|---|
| $h^*$ | 1st | 2nd | $\mathcal{H}^1 \to \mathcal{H}^1$ | $\mathcal{H}^1 \to \mathcal{H}^2$ | $\mathcal{H}^2 \to \mathcal{H}^1$ | $\mathcal{H}^2 \to \mathcal{H}^2$ |
| $\mathcal{H}^1$ | (1/2) | (1/2) | 0.54 | 0.13 | 0.06 | 0.27 |
| $\mathcal{H}^2$ | (1/2) | (1/2) | 0.30 | 0.31 | 0.01 | 0.38 |
| $\mathcal{H}^2$ | (1/2) | 2 | 0.00 | 0.61 | 0.00 | 0.39 |
| $\mathcal{H}^2$ | 2 | 2 | 0.00 | 0.00 | 0.00 | 1.00 |
| $\mathcal{H}^2$ | 2 | (1/2) | 0.00 | 0.00 | 0.11 | 0.89 |

(a) Transitions across subclasses $\mathcal{H}^i \to \mathcal{H}^j$      (b) Transitions within $\mathcal{H}^1$     (c) Teaching results

Figure 4: (a)-(b) represent results for eliciting human preferences for different session types as explained in the text below and (c) shows results for teaching human learners. (a) Participants prefer staying within the same hypothesis subclass when possible, displayed as the fraction of time they switched subclasses for different session types. (b) Considering the transitions within subclass $\mathcal{H}^1$, participants favor staying at their current hypothesis if it remains valid, along with preferring smaller updates, computed as the $L1$ distance between the initial and updated rectangle. (c) Adaptive teaching algorithm Ada-R is significantly better than SC and Rand.

that participants tend to favor staying in the same hypothesis subclass when possible. Within the same subclass, they have a preference towards updates that are close to their initial hypothesis, cf. Fig. 4b.[5]

**Teaching human learners** Next we evaluate our teaching algorithms on human learners. As in the simulations, we consider four teaching scenarios $\mathcal{H}^{1\to1}$, $\mathcal{H}^{1\to2}$, $\mathcal{H}^{2\to1}$, and $\mathcal{H}^{2\to2}$. At the beginning of the teaching session, a participant was shown a blank $8 \times 8$ grid with either one or two initial rectangles, corresponding to $h^0$. At every iteration, the participants were provided with a new teaching example (i.e., a new green or blue cell is revealed), and were asked to update the current hypothesis.

We evaluate three algorithms, namely Ada-R, SC, and Rand, where Rand denotes a teaching strategy that picks examples at random. The non-adaptive algorithm Non-R was not included in the user study for the same reasons as explained in Footnote 3. We enlisted 200 participants to evaluate teaching algorithms and this was repeated five times for each participant. For each trial, we randomly selected one of the three teaching algorithms and one of the four teaching scenarios. Then, we recorded the number of examples required to learn the target hypothesis. Teaching was terminated when 60% of the cells were revealed. If the learner did not reach the target hypothesis by this time we set the number of teaching examples to this upper limit. We illustrate a teaching session in the extended version of this paper [8].

Fig. 4c illustrates the superiority of the adaptive teacher Ada-R, while Rand performs the worst. In both cases where the target hypothesis is in $\mathcal{H}^2$, the SC teacher performs nearly as well as the adaptive teacher, as at most 12 teaching examples are required to fully characterize the location of both rectangles. However, we observe a large gain from the adaptive teacher for the scenario $\mathcal{H}^{2\to1}$.

# 8 Conclusions

We explored the role of adaptivity in algorithmic machine teaching and showed that the adaptivity gain is zero when considering well-studied learner models (e.g., "worst-case" and "preference-based") for the case of version space learners. This is in stark contrast to real-life scenarios where adaptivity is an important ingredient for effective teaching. We highlighted the importance of local preferences (i.e., dependent on the current hypothesis) when the learner transitions to the next hypothesis. We presented hypotheses classes where such local preferences arise naturally, given that machines and humans have a tendency to learn incrementally. Furthermore, we characterized the structure of optimal adaptive teaching algorithms, designed near-optimal general purpose and application-specific adaptive algorithms, and validated these algorithms via simulation and user studies.

**Acknowledgments** This work was supported in part by Northrop Grumman, Bloomberg, AWS Research Credits, Google as part of the Visipedia project, and a Swiss NSF Early Mobility Postdoctoral Fellowship.

## Footnotes

[1]For simplicity of discussion, we assume that for the 2-REC hypothesis that contains two rectangles, the edges of the two rectangles do not overlap.

[2]In the case of ties, we assume that the teacher prefers examples that make learner stay at the same hypothesis.

[3] In general, the teaching sequence constructed by the non-adaptive algorithms Non-R (resp. Non-L) would not be sufficient to reach the target under the noisy setting. Hence, we did not include the results of these non-adaptive algorithms in the robustness plots. Note that one can tweak Non-R (resp. Non-L) by concatenating the teaching sequence with teaching examples generated by SC; however, in general, in a worst-case sense, any non-adaptive algorithm in the noisy setting will not perform better than SC.

[4] The performance of SC is non-monotone w.r.t. the noise-level. This is attributed to the stopping criteria of the algorithm as the increase in the noise level increases the chance for the learner to randomly jump to $h^*$.

[5]Given that a participant is allowed to move edges when updating the hypothesis, our interface could bias the participants' choice of the next hypothesis towards a preference structure that favors local edits as assumed by our algorithm. As future work, one could consider an alternative interface which enforces participants to draw the rectangle(s) from scratch at every step.

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
