[Reviews · NeurIPS 2018]

Reviewer 1



This paper considers when knowing a learner's current hypothesis can lead to more effective teaching than if the hypothesis is not known. The benefits of adaptivity are explored in two cases where there is an impact, and algorithms are developed for adaptive teaching in those domains that attain efficient performance. The growth rate of the number of examples for adaptive and non-adaptive teachers is shown, and adaptive algorithms are compared to alternatives both in simulation and with human learners. This paper addresses an important issue in algorithmic teaching, as adaptivity is often costly and understanding when adaptivity is important could be useful for building adaptive teaching systems. I appreciate that the paper made clear conceptually, as well as mathematically, when adaptivity matters. The paper focuses on local preference updating, often making reference to cases where the hypothesis does change but changes in a way that's dependent on the current hypothesis, but my reading is that any dependence on the current hypothesis has the potential to have adaptivity matter, including simply having the hypothesis stay the same if new data doesn't violate it. I bring up this case because it's consistent with some simple human learning models, such as the "win stay lose shift" model (e.g., Levine, M. (1975). A cognitive theory of learning: Research on hypothesis testing; Bonawitz, E., Denison, S., Gopnik, A., & Griffiths, T. L. (2014). Win-Stay, Lose-Sample: A simple sequential algorithm for approximating Bayesian inference. Cognitive psychology, 74, 35-65; Rafferty, A. N., Brunskill, E., Griffiths, T. L., & Shafto, P. (2016). Faster teaching via POMDP planning. Cognitive science, 40(6), 1290-1332). It would be interesting to explore how much adaptivity is helpful with such a simple model that is less constrained in what happens when the current hypothesis is violated - the preference function \sigma is essentially local only when the current hypothesis is still valid. Making links to this type of work with human learning models could strengthen the impact of the present paper. The main questions I had center on the human evaluation, where I believe the results are overstated. The "Real-world Human Preferences" uses data from participants to argue that people perform incremental updates when learning in 2-Rec. While I agree this is probably the case, the section doesn't acknowledge the (likely large) impact of the interface on these results. The participant has a hypothesis shown on screen at all times, and I believe participants were asked to "update" this based on the new information. This interface constrains people to match the version space learner (at least in terms of what we can observe) by having exactly one hypothesis, and that hypothesis must be consistent with all data provided so far; while other representations are possible and have been frequently used for representing human category learning (e.g., maintaining an implicit distribution over possible hypotheses), these can't be expressed. The interface also nudges people towards local updates - e.g., I expect results would be at least somewhat different if people had to draw new rectangles every time, on a different image than the one with the hypothesis displayed (and probably more different still if they couldn't see the previously drawn hypothesis). Thus, the human evaluation seems to show that people are helped by an adaptive algorithm (as compared to random examples or set cover), given an interface that pushes them towards the learner model that is assumed by the adaptive algorithm. I still think this is important and useful to show, but the limitations of this evaluation should be acknowledged. Additional questions about the human evaluation: - Why wasn't the non-adaptive algorithm (non-R) considered? That seems like it would be the natural comparison for understanding how much adaptivity matters in this case, and I would very much like to see results for the non-R case, more closely mirroring the simulated case. - I did not understand the different between the target hypothesis being M_{1} versus M_{1/2} in Figure 3a. My understanding from the other cases where the subscript 1/2 is used is that it is for cases where either 1 or 2 rectangles could be consistent with the data. Is the distinction here that M_{1} is a single square (so non-representable using a 2 rectangle hypothesis), but M_{(1/2)} is a larger rectangle, which could be expressed by 1 rectangle or by 2 adjacent rectangles? If so, why this case is separated out for the human learners and not for the simulated evaluation? I may very well be missing something and clarification would be appreciated. The consideration of noise in the preference function in the supplement was helpful in thinking about applications of this work. In the noisy case, why were only SC and Ada considered, not Non? As above, Ada versus Non seems like the most interesting comparison to see the impact of adaptivity with both adaptive and non adaptive having the (noisily) correct preference function. I look forward to the authors making the point about SC outperforming Non and incorporating the additional baseline, as discussed in their response. To summarize based on the four review criteria: - Quality: The theoretical work here is sound, and the simulations support these theoretical results. The behavioral experiment is a key part of supporting the relevance of this work for human teaching, and it provides moderate support for the relevance of the theoretical work, but as discussed above, the limitations of the current design and how it drives people's behavior towards the proposed model are not adequately discussed. - Clarity: The paper is well-organized and understandable, with an appropriate level of detail in both the main paper and the supplementary materials. There are several individual points that were not clear (my questions are provided elsewere in this review), but as a whole, clarity is good. - Originality: To the best of my knowledge, framing the adaptivity question in this rigorous theoretical way is novel, and the work appropriately cites related work and argues for why this work adds to the existing literature. - Significance: The paper goes beyond previous work, exploring the importance of adaptivity from three methodologies (theoretical, simulation, and behavioral experiments), and is likely to be of interest both to those interested specifically in the theoretical results for machine teaching and those who are more oriented towards educational applications. Other researchers in machine teaching are likely to build on this work, and the rigorous approach of determining when adaptivity makes teaching/learning more efficient is likely to be relevant to applied work in using machine learning approaches for education. After reading the author response and the other reviews, my general evaluation of the paper remains the same. The response clarifies several questions, and the proposed revisions will, I believe, increase the clarity of the paper. While computing the optimal teaching policy is hard (and that problem is not solved by this work), I think it is an important contribution to formalize the problem in this way, and will perhaps facilitate future work that focuses on techniques for the optimization problem. I'm unsure from the response if the new experimental study is intended to be included in this work or in future work; if the latter, I again want to strongly urge the authors to note the limitation of how the present interface may encourage participants towards the assumed learner model. Small issues/typos: - Line 10: "learner picks hypothesis according to..." -> hypotheses - Line 241: "poorly.An" - Line 329: "an web-interface" - Line 348: "There were randomly assigned" - Caption Figure 3: "require less examples" -> "require fewer examples" - Line 397 in the supplement: "100 participants on Mechanical Turk were each shown the same set of then initial configurations." -> ten, not then

Reviewer 2



Summary: This paper considers the problem of machine teaching when the learner has a general type of preference function over hypotheses, where the amount a learner prefers a particular hypothesis depends on its current hypothesis. Specifically, they are interested in understanding when an iterative/adaptive teaching protocol fairs better than a non-adaptive approach. They demonstrate that when a learner only has global preferences (i.e. their preferences do not depend on their current hypothesis), there is no gain in an adaptive strategy. They also provide two example settings in which one can show adaptivity can lead to improvement over the non-adaptive setting. They also present conditions under which a certain greedy strategy is within some logarithmic factor of optimal. Quality: This paper appears to be technically sound, although I have not checked all of the proofs rigorously. Clarity: This paper is reasonably clear, but I did have difficulty following the discussions surrounding the example classes (2-Rec and Lattice). For example, the discussion in the first paragraph of Section 5.2 introduces an oracle S(), a subroutine Teacher(), and a term “sub-task” that all seem to be interrelated but don’t seem to be formally defined. Originality + significance: The local preference model of a learner in the machine teaching setting appears to be an interesting contribution to the field. It is well-motivated and very general. Unfortunately, the algorithms considered here only seem to be feasible in relatively toy settings. In particular, to implement Algorithm 1, we need to find the hypothesis which maximizes \tilde{D}(h). However, this appears to require enumerating over the entire candidate set (which may require enumerating the entire version space to construct). This paper would be strengthened significantly by a reasonable example where (say approximately) maximizing \tilde{D}(h) can be done efficiently. ———————————————————————————————————— Revision after author response: After viewing the author response and the other reviews, I have decided to revise my score upwards. My main objection to the current work was that the proposed algorithms only work in toy settings due to the difficulty of the optimization problems. However, as pointed out by another reviewer, this paper has several important contributions, and the formalisms presented here open the door for others to work in this area.

Reviewer 3



This paper addresses the following question in machine teaching: When does an adaptive teacher outperforms a non-adaptive teacher? Adaptivity refers to the teacher’s ability of picking the examples in an adaptive manner; namely the next example to be presented to the learner may depend on the current state of the learner (in contrast, a non-adaptive teacher presents a batch of examples that depend only on the initial state of the learner). Main results: 1. Formalize of a mathematical framework in which this question can be studied quantitatively (and showing that previously studied models fit in as special instantiations in the new model) 2. Provide explicit examples of teaching tasks (Examples 1 and 2 in the paper) in which adaptivity helps (Theorem 2), and showing that for a previously studied class of learners adaptivity does not help (Proposition 1). 3. Design a heuristic teaching algorithm (Algorithm 1) and state sufficient conditions under which this algorithm attains provable guarantees (Theorem 4) The paper well written and the presented framework is natural and I believe it would be interesting to the ML community.